# Impact of Surface Roughness on Flow Physics and Entropy Generation in Jet Impingement Applications

**DOI:** 10.3390/e24050661

**Published:** 2022-05-08

**Authors:** Abdulrahman Alenezi, Abdulrahman Almutairi, Hamad Alhajeri, Saad F. Almekmesh, Bashar B. Alzuwayer

**Affiliations:** 1Department of Mechanical Power and Refrigeration, College of Technological Studies, P.AA.E.T., Kuwait City 70654, Kuwait; asa.almutairi@paaet.edu.kw (A.A.); hm.alhajeri@paaet.edu.kw (H.A.); sf.almokmesh@paaet.edu.kw (S.F.A.); 2Department of Automotive and Marine Engineering Technology, College of Technological Studies, P.A.A.E.T., Kuwait City 70654, Kuwait; bb.alzuwayer@paaet.edu.kw

**Keywords:** entropy, heat transfer, CFD, jet impingement, rib

## Abstract

In this paper, a numerical investigation was performed of an air jet incident that normally occurs on a horizontal heated plane. Analysis of flow physics and entropy generation due to heat and friction is included using a simple easy-to-manufacture, surface roughening element: a circular rib concentric with the air jet. This study shows how varying the locations and dimensions of the rib can deliver a favorable trade-off between entropy generation and flow parameters, such as vortex generation and heat transfer. The performance of the roughness element was tested at three different radii; R/D = 1, 1.5 and 2, where D was the jet hydraulic diameter and R was the radial distance from the geometric center. At each location, the normalized rib height (e/D) was increased from 0.019 to 0.074 based on an increment of (e/D) = 0.019. The jet-to-target distance was H/D = 6 and the jet Reynolds number (Re) ranged from 10,000 to 50,000 Re, which was obtained from the jet hydraulic diameter (D), and the jet exit velocity (U). All results are presented in the form of entropy generation due to friction and heat exchange, as well as the total entropy generated. A detailed comparison of flow physics is presented for all ribs and compared with the baseline case of a smooth surface. The results show that at higher Reynolds numbers, adding a rib of a suitable height reduced the total entropy ($S_{t}$) by 31% compared to the no rib case. In addition, with ribs of heights 0.019, 0.037 and 0.054, respectively, the entropy generated by friction ($S_{f}$) was greater than that due to heat exchange ($S_{h}$) by about 42%, 26% and 4%, respectively. The rib of height e/D = 0.074 produced the minimum *S_t_* at R/D = 1. As for varying R/D, varying rib location and Re values had a noticeable impact on $S_{h}$, $S_{f}$ and ($S_{t}$). Placing the rib at R/D = 1 gave the highest total entropy generation ($S_{t}$) followed by R/D = 1.5 for all Re. Finally, the Bejan number increased as both rib height and rib location increased.

## 1. Introduction

In recent years, the concept of minimizing all types of entropy generation in jet impingement applications has been a major concern of engineers and scientists in many different fields. Bejan [1,2] was the first to study the sources of different types of entropy generated in engineering processes and quantify them. Ruocco [3] studied entropy generation when a cooling laminar air-jet impinged on a horizontal heated plate. He found that the integrated entropy generation rate increased proportionally with the thermal conductivity ratio where thermal conductivity had no influence on the average Nusselt number (Nu).

Flow physics, heat transfer and entropy generation on roughened surfaces have long been a topic of interest among scientists and have many different industrial applications. A better understanding of the physics of flows and the transfer of heat to and from flat surfaces with roughening elements is an ongoing requirement in many industrial fields.

An early study on entropy generation by Shuja et al. [4], tested various turbulence models to calculate the rate of generation of entropy of a jet impinging onto a heated plane. Their major finding was that entropy generation occurred in two main regions: close to the stagnation region at radial distances between 0 ≤ R/D ≤ 1, and the shear layer between the jet and the ambient air. Shuja et al. [5,6] continued to study parameters that influence entropy generation and heat transfer, such as inlet velocity profiles on an adiabatic wall impinged by laminar swirling jets. The authors found that entropy generation was at its maximum across the jet outer boundary compared to other locations. This may have been because higher temperature gradients exist along jet outer boundaries. It was also reported that as swirl velocity increases, so do the irreversibilities due to enhanced heat exchange, while the entropy generated by fluid friction decreases. Increasingly, the generation of entropy is included in the analyses of impingement heat transfer [7,8,9]. Xu et al. [10] investigated the effect of surface roughness on Nu and heat transfer distribution, using the second law of thermodynamics. They sought to optimize the heat transfer of impinging jets by minimizing the generation of entropy. The latter study also included a numerical analysis of the effect of employing a 2D wavy surface on entropy and heat transfer rate. The authors reported a slight enhancement in the average rate of heat transfer of a wavy surface when evaluated against a smooth one, and suggested an optimal design for jet impingement to minimize entropy generation. Esmailpour et al. [11] explored entropy produced in thermal and flow fields generated by a pulsating impinging jet. Their study highlighted that the entropy generated increased as the nozzle-to-plate distance decreased, with maximum thermal entropy generated when the jet was closest to the flat surface.

Zahmatkesh [12] studied how conditions at the thermal boundary affected both entropy generation and heat transfer for uniform/non-uniform heated and cooled surfaces. The author demonstrated that flow physics, heat transfer and entropy generation inside a closed space can all be significantly affected by thermal boundary conditions, with the optimum entropy generation and heat transfer occurring under conditions of non-uniform heating. Ries et al. [13] studied (experimentally) the heat transfer and fluid flow characteristics of an air jet striking an inclined flat plane. Where the jet flow was directed downwards (vertically) onto a horizontal plane, they found the maximum Nu occurred at the stagnation point. Chaudhari et al. [14] measured the pressure and temperature of jet flow impinging on a plate to determine how heat transfer and pressure drop varied with radial distance. The tests were limited to Reynolds numbers between 1500 and 4200, and the axial distance between the jet and heated plate, with the jet limited to 0 to 25 jet diameters.

When analyzing entropy generation using CFD software version 18.1 (ANSYS, Inc., Pennsylvania, USA), it is common practice to use the local form of the second law of thermodynamics when investigating thermodynamic irreversibilities. This permits the total entropy generated in a system to be quantified and to investigate how irreversibilities are dispersed locally in the system [15]. Using the notion of minimal entropy generation, CFD analyses can identify sources of irreversibilities for an extensive range of thermo-fluid processes including laminar and turbulent heat transfer in wall-bounded flows [16,17,18,19,20,21,22], reacting flows [23,24] and heat transfer with impinging flows [25]. The contributions made to the analysis and application of entropy generation using CFD software for a range of engineering systems are reviewed in [26].

A numerical study of the generation of entropy, as a result of the combined effects of a nanofluid within a heated cavity which had porous fins attached to its surfaces, was presented by [27], based on a nanoparticle percentage volume fraction between 0–4% under the influence of multiple ranges of both the Darcy number and Rayleigh number. The authors reported that for the lower Rayleigh numbers, higher rates of entropy generation were recorded because of minimum convective heat transfer due to a high-temperature gradient along the fins. On the other hand, enhanced flow circulation in the channels and an increased Rayleigh number led to an increase in both flow velocity gradient and entropy generation.

An investigation by Chen and Zheng [28] using the Lattice Boltzmann Method (LBM) examined the effect on entropy generation of using planar opposing jets in jet impingement applications. The authors focused in particular on two jet impingement parameters: Reynolds number and jet-to-target distance, showing that the distance ratio between opposing jets (W/L) has more influence on entropy generation than the Reynolds number. Lam and Prakash [29] numerically investigated the effect of a porous medium on heat transfer and entropy generated during jet impingement on protruding heat sources. They suggested that a porous layer should be attached to the impingent plate to reduce hot spots which resulted from recirculation bubbles and suggested an optimal configuration to give maximum overall heat transfer combined with the least entropy generation.

In a recent study by Salimi et al. [30], the authors developed a 3D numerical model to incorporate features of both heat sink and entropy generation of jet flow impinging on a porous surface, assuming local thermal non-equilibrium constrains. They found that the thickness of the porous layer and the Reynolds number of the jet had more effect on Nu and average viscous entropy generation than the other factors considered.

Alenezi et al. [31] examined the heat transfer of a cooling jet impinging on a hot surface, as well as considering how varying the Reynolds number, roughness location and roughness dimensions affected the heat transfer and flow physics parameters, including average heat transfer coefficient, wall shear and flow vortices with thermodynamic non-equilibrium. The authors also examined how varying both dimensions and location of the roughness element, which was in the shape of a continuous rib of square cross-section, provided a favorable trade-off between the rate of heat transfer and the total pressure loss. The results demonstrated that changes to the size and the geometry of the roughening elements could significantly affect the average heat transfer coefficient. Depending on the precise changes made, the heat transfer could be either increased or decreased relative to the baseline. The average heat transfer coefficient was increased by almost 13% for the optimum dimensions and location of the roughness element for a given Reynolds number. However, the average heat transfer coefficient decreased from 10% to 30% when a rib height of 0.074 was used at a Reynolds number of 50,000. Varying either rib location and/or rib height could have improved heat transfer for low Reynolds numbers.

This paper aims to progress previous research [31] by examining (numerically) the effects of varying both the location and dimensions of a roughening element (rib) protruding from a horizontal surface on entropy generation and flow physics, to deliver a favorable trade-off between them. A detailed analysis of flow physics (i.e., pressure, wall shear and flow vortices) and entropy generation (i.e., friction, heat, and total entropy) for different locations of the ribs for an air jet impinging on an isothermal flat plate for various Reynolds numbers is performed. In addition, the effect of different rib geometries at each location on entropy generation will be investigated to achieve optimal rib configuration.

## 2. Computational Setup and Numerical Scheme

### 2.1. Computational Domain and Assumptions

Figure 1 shows the computational domain, mesh details and boundary conditions for the proposed problem. Due to the axisymmetric nature of the flow and geometry, only half of the domain was simulated to save computational time. All dimension details were extracted from the experimental work of O’Donovan and Murray [32] and were normalized to the jet hydraulic diameter (D) to validate the current study. The domain, which has a circular shape, has a diameter of 40 D and includes the isothermally heated wall. The air jet, which has a fully developed profile, was separately pre-simulated and imposed at the jet inflow. Table 1 shows all the air properties used in this simulation.

A very fine mesh was used on the whole domain, especially for the stagnation region and the rib, to accurately capture the heat transfer rate. The wall y+ is a dimensionless number defined as the distance between the wall and the first mesh above. The y+ value near the wall was kept below one, as recommended in the literature for these types of jet impingement applications. The fluid used here is air at room temperature leaving an unconfined jet with a hydraulic diameter of 13.5 mm to impinge on an isothermal flat surface (wall), which was maintained at 60°. The wall has a no-slip condition and was located at a distance H/D = 6 vertically below the jet. The top and side surfaces of the numerical domain were set to be pressure outlets maintained at ambient temperature. Reynolds numbers were set to values between 10,000 and 50,000. All air properties were for 1 atm and room temperature. Finally, the numerical results were successfully validated against the experimental data of O’Donovan and Murray [32], which has a smooth surface (no rib) and will be referred to as the baseline in all later calculations.

### 2.2. Governing Equations

The three-dimensional steady-state incompressible Reynolds Average Navier–Stokes (RANS):(1)$$\frac{\partial U}{\partial Z}+\frac{\partial U}{\partial r}+\frac{V}{r}=0$$
(2)$$\rho \left(U\frac{\partial U}{\partial Z}+V\frac{\partial U}{\partial r}\right)=-\frac{\partial P}{\partial Z}+\frac{1}{r}\frac{\partial }{\partial r}\;\left[r\left(\mu \frac{\partial U}{\partial r}-\rho \bar{u^{\prime }v^{\prime }}\right)\right]+\frac{\partial }{\partial z}\left[\left(\mu \frac{\partial U}{\partial z}-\rho \bar{u^{\prime }u^{\prime }}\right)\right]$$
(3)$$\rho \left(U\frac{\partial V}{\partial Z}+V\frac{\partial V}{\partial r}\right)=\frac{-\partial P}{\partial r}+\frac{1}{r}\frac{\partial }{\partial r}\left[r\left(\mu \frac{\partial V}{\partial r}-\rho \bar{v^{\prime }v^{\prime }}\right)\right]+\frac{\partial }{\partial z}\left[\left(\mu \frac{\partial U}{\partial z}-\rho \bar{u^{\prime }u^{\prime }}\right)\right]-\mu \frac{V}{r^{2}}$$
(4)$$\rho C_{p}\left(U\frac{\partial T}{\partial Z}+V\frac{\partial T}{\partial r}\right)=\lambda \left[\frac{\partial }{\partial r}\left(\frac{\partial T}{\partial r}\right)+\frac{1}{r}\frac{\partial T}{\partial r}+\frac{\partial }{\partial z}\left(\frac{\partial T}{\partial z}\right)\right]+\bar{\Phi }-\rho C_{p}\left(\frac{\partial \bar{u^{\prime }t^{\prime }}}{\partial z}+\frac{\partial \bar{v^{\prime }t^{\prime }}}{\partial r}-\frac{\partial \bar{v^{\prime }t^{\prime }}}{r}\right)$$
where $\bar{\Phi }$ is the viscous dissipation heat source
(5)$$\bar{\Phi }=2\;\mu \;[{\left(\frac{\partial V}{\partial r}\right)}^{2}+{\left(\frac{V}{r}\right)}^{2}+\frac{1}{2}{\left(\frac{\partial V}{\partial z}+\frac{\partial u}{\partial z}\right)}^{2}]$$

In the example highlighted above, ρ, *P*, and *T* are the density, average pressure, and temperature, respectively. *U* and *V* are velocity components in the *z* and *r* directions, whereas $u^{\prime }$, $v^{\prime }$, and $t^{\prime }$ are the fluctuating velocity components and temperature in the *z* and *r* directions. Finally, *C_p_* is the heat capacity of air at constant pressure.

A *k*-*ε* model derived by RNG is studied numerically in Martinelli-Yakhot [33]. For incompressible flows, the model is:
(6)$$\partial_{t}k+u\nabla k-\frac{\nu_{T}}{2}{\left|\nabla u+\nabla u^{T}\right|}^{2}+\varepsilon -\nabla \cdot (\alpha \nu \nabla k)=0\;$$
(7)$$\partial_{t}\varepsilon +u\nabla \varepsilon -\frac{\nu_{T}}{2}\;\sqrt{\varepsilon }C_{c}Y_{\varepsilon }{\left|\nabla u+\nabla u^{T}\right|}^{2}+\varepsilon^{\frac{3}{2}}Y_{\varepsilon }-\;\nabla \cdot (\alpha \nu \nabla \varepsilon )=0$$
with $C_{c}$ ≈ 75. The eddy viscosity $\nu_{T}$ is related to $k^{2}/\varepsilon$ via a differential equation and inverting the result
(8)$$\frac{dY_{\varepsilon }}{d_{\nu_{T}}}=-0.5764{\left(\nu_{T}{}^{3}+C_{c}-1\right)}^{-\frac{1}{2}}$$

The function *α* depends upon $\nu_{T}$ by
(9)$${\left(\frac{1.3929-\alpha )}{0.3923}\right)}^{0.63}{\left(\frac{2.3929-\alpha }{3.3929}\right)}^{0.37}=\nu_{T}{}^{-1}$$

The model contains its own low-Reynolds number version, and at a high Reynolds number it gives the usual law $\nu_{T}=c_{\mu }k^{2}/\varepsilon$ but with $c_{\mu }=0.084$.

The total entropy generation rate, $(S_{t}$), can be calculated using:
*S_total_* (*S_t_*) = *S_heat_* (*S_h_*) + *S_friction_* (*S_f_*)(10)
where *S_h_* is entropy generated from heat exchange between flow and heated wall and *S_f_* is due to friction occurring between flow and heated wall Kock and Herwig [34].

*S_h_* and *S_f_* as:(11)$$S_{h}=\frac{\lambda_{eff}}{T_{i}^{2}}\;[{\left(\frac{\partial T}{\partial x}\right)}^{2}+{\left(\frac{\partial T}{\partial y}\right)}^{2}+{\left(\frac{\partial T}{\partial z}\right)}^{2}]\;$$
where $\lambda_{eff}$ is the effective thermal conductivities of the flows, laminar and turbulent.
(12)$$S_{f}=\frac{\mu_{eff}}{T_{i}}\{2[{\left(\frac{\partial u}{\partial x}\right)}^{2}+{\left(\frac{\partial v}{\partial y}\right)}^{2}+{\left(\frac{\partial w}{\partial z}\right)}^{2}]+{\left(\left(\;\frac{\partial v}{\partial x}\right)+\left(\;\frac{\partial u}{\partial x}\right)\right)}^{2}+{\left(\left(\;\frac{\partial u}{\partial z}\right)+\left(\;\frac{\partial w}{\partial x}\right)\right)}^{2}+{\left(\left(\;\frac{\partial v}{\partial z}\right)+\left(\;\frac{\partial w}{\partial y}\right)\right)}^{2}\}$$
where $\mu_{eff}$ is the effective viscosity of the flows, laminar and turbulent.
*μ**_eff_* = *μ**_laminar_* + *μ**_turbulent_*(13)

The Bejan number is the ratio between entropy generated due to heat transfer and total energy generated.
(14)$$\mathrm{Bejan}\;\mathrm{number}=\frac{S_{h}}{S_{t}}$$

The average entropy generation due to heat transfer and fluid friction over a circular domain with radius *R* can be obtained by:(15)$$\bar{S_{h,R}}=\int_{0}^{R}S_{heat}\;\mathrm{dR}$$
(16)$$\bar{S_{f,R}}=\int_{0}^{R}S_{friction}\;\mathrm{dR}$$

### 2.3. Modeling Approach and Validation

Axisymmetric jet impingement was simulated using ANSYS simulation software version 18.1. An initial study was performed to investigate which of the three grid sizes to use, to determine the one that offered adequate accuracy for the least computational time. Special care was taken when considering the near-wall region as this is important for convective heat transfer.

The SIMPLEC scheme, which uses the momentum and continuity equations, was combined with a Green–Gauss Cell-Based solver, which provides the gradients of the required variables at the centers of the cells for spatial discretization. Second-order discretization was used to solve the momentum and energy equations to obtain more accurate results, while first-order discretization was considered satisfactory for solving the remaining equations. Before accepting a solution as the most appropriate, the entire domain was initialized according to the inlet flow conditions, using first-order upwind discretization to obtain a convergence criterion at 10^−6^ for the energy equation and 10^−4^ for the remainder. A second step combined different orders of discretization to find a solution.

Figure 2 shows results for an impingement jet angle of 90° and a jet to plane distance of H/D = 6. We see that the local value of the simulated Nu decreased with radial distance (R/D) but did not vary between mesh sizes and was close to the experimentally obtained values. To minimize run times, the 400,000 grid was used for the remaining simulations.

Trial runs to compare measured values of the local Nu using the RNG k-epsilon, SST k-omega, and the Reynolds stress turbulence models were carried out to assess the models’ relative accuracy of prediction. The measured values of Nu were obtained from O’Donovan and Murray [32] and Alimohammadi [35] for the same physical arrangement and experimental conditions. The RNG k-epsilon showed the best overall agreement with the experimental data and was able to predict the local stagnation point Nusselt number (𝑁𝑢_𝑠𝑡𝑎𝑔_) with an error of less than 2%. However, both RSM and SST k-omega overvalued 𝑁𝑢_𝑠𝑡𝑎𝑔_, the first by 18% and the latter by 21%. Moving radially away from the jet, the RNG k-omega provided (overall) a more accurate prediction of local Nu values. However, neither model was able to discern the small trough seen at R/D ~ 2 in the experimental results, which produced a small peak at R/D ~ 3. Based on this evaluation, the RNG k-epsilon turbulence model was chosen for this parametric study.

## 3. Results and Discussion

### 3.1. Smooth Wall—The Baseline Case

Flow Physics Analysis

The flow regime can be specified by the value of Reynolds number (Re), which is defined as the ratio of inertia to viscous forces. Based on the bulk jet exit velocity (U) and nozzle diameter (D), it can be defined as:(17)Re=ρUDμ=UDν
where μ and ν are the dynamic and kinematic viscosities of the fluid, respectively.

Shear stress at the wall is defined by:(18)$$\tau_{w}=\left(\mu +\mu_{t}\right)\frac{\partial u}{\partial n}$$

Figure 3 shows local and average wall shear stress ($\tau_{w})$ distribution in the downstream direction for different Reynolds numbers. The presence of a transverse vortex when the flow hits the plate is responsible for flow expansion, which results in a separation between the flow and the impingement wall. Thus, the wall shear stress is highly reduced at the stagnation point. It should be noted that the peak wall shear stress is achieved at a location between R/D = 0.7 and 0.8, where the interaction between the wall and primary vortex occurs resulting in a secondary vortex generation. This peak value usually corresponds to the highest flow velocity (U) value maximum pressure gradient (*∂P*/*∂y*). Past this point, a gradual decrease in shear stress continues until it almost reaches a constant value at radial distance R/D ≥ 9 due to the negligible pressure gradient at R/D ≥ = 1.5. Overall, the Reynolds number has a major effect on shear stress beyond R/D = 0. Comparing the shear stress peak value difference between the minimum and maximum Reynolds number, a 564% percentage increase can be observed. The area-average wall shear stress ($\bar{\tau_{w}})$ for different values of Reynolds numbers over an area with a diameter = 2.5 D is shown in Figure 3b. The maximum value of average wall shear stress at the highest Reynolds number is 2700% of its value at the lowest Re.

Figure 4 presents turbulent intensity (I) on the impingement surface with a changing Reynolds number for the range 0 ≤ R/D ≤ 6. Figure 4a shows that the turbulence intensity values at the stagnation point increased slightly with an increase in the Reynolds number due to the near-wall effect on the flow. As the flow traveled away from the stagnation region, turbulence intensity increased sharply, especially for higher Reynolds numbers, until it reached a maximum value at radial location R/D = 0.8 for Re = 50,000. This was due to the presence of different sizes of vortices which enhanced the flow turbulence. Beyond this point, a sharp decrease in turbulence intensity was observed due to the momentum exchange between the wall and the flow, until radial distance R/D = 4.5, after which it maintained a constant value for Reynolds numbers between 4 ≥ R/D ≥ 6. The area-averaged turbulence intensity ($\bar{I}$) for different values of Reynolds numbers over an area with outer diameter = 2.5 D is shown in Figure 4b. Compared to its value at Re = 10,000, the value of the average turbulence intensity at Re = 50,000 has increased by about 1840%.

b.Entropy generation analysis

Figure 5 shows entropy generation due to friction ($S_{f}$), due to heat exchange ($S_{h})$ and total entropy generated ($S_{t}$) for different Re at different radial locations (R/D). Entropy due to friction alone has a zero value at the stagnation point, unlike entropy due to heat transfer between the heated plate and the flow, which has a maximum value at this point because of the high heat exchange between the heated wall and the stagnant fluid.

As the flow moves downstream,$\;S_{f}$ increased sharply and almost linearly, until it reached a maximum value within the stagnation region at radial location R/D ≈ 0.8, due to momentum exchange between the moving fluid and heated wall. Figure 5a clearly shows that the rate of increase of $S_{f}$ was highly dependent on Re. It can be observed that Re had a significant effect on both $S_{f}$ and $S_{h}$ within the stagnation region, where most of the large vortices exist, resulting in a very active flow.

Figure 6 shows the relationship between entropy due to friction (*S_f_*) and entropy due to heat exchange (*S_h_*) with flow turbulence intensity (I) and wall shear ($\tau_{w})$ for different radial distances (R/D) for a mid-range flow speed, Re = 30,000. The figure demonstrated that both turbulence intensity (I) and *S_f_* increase with radial distance, and the combination showed a maximum at R/D ≈ 0.75. This was due to the high flow velocity at this radial location and the existence of a secondary vortex generation.

It is clear that the rate of increase and decrease *S_f_*, as a function of R/D, was much sharper than for either turbulence intensity or wall shear. For example, when compared to peak values, the percentage decrease at R/D = 3 was about 92% for *S_f_* and 52% for I. The plots for $\tau_{w}$ show the same trend, where the percentage decrease at R/D = 3 was about 92% for *S_f_* and 72% for $\tau_{w}$, when compared to peak values. The relationship between entropy due to heat exchange, *S_h_*, and flow turbulence intensity and wall shear are shown in the lower panels of Figure 6.

### 3.2. Roughened Wall—Rib Geometry Effect

Flow Physics Analysis

Figure 7 shows how varying rib height (e/D) affects wall shear stress $\tau_{w}$ for two values of the Reynolds number, 20,000 and 40,000, located at R/D = 1.5. For Re = 20,000, the presence of any of the ribs increased local wall shear values, above that of the baseline by approximately 23% over the range R/D = 0 to R/D ≈ 1.2. However, for Re = 40,000, the presence of a rib made no significant change to the local wall shear values over the same R/D range. In general, wall shear stress to the downstream distance from the stagnation point as the flow started to accelerate.

Between R/D ≈ 1.2 and the rib, the presence of the rib affected the flow by lowering the local wall shear values to almost zero, where flow recirculation occurred before the rib, preventing direct contact between the flow and the wall. When the flow meets with the rib, a substantial jump to a peak value of shear stress at the face of the rib was observed. The magnitude of this peak depends mainly on the rib height, and the peak is located on the top front face of the rib, where the flow impacted the rib, as shown in the contour. The Figure also showed that e/D = 0.019 produced the greatest wall shear stress peak value with percentage increases of about 875% and 600% compared to the baseline case for Re = 20,000 and 40,000 respectively. The Figure shows that a rib of height e/D = 0.074 produced the lowest peak value for both Reynolds numbers.

Figure 8 shows local turbulence intensity variation on the heated surface for Reynolds numbers of 20,000 and 40,000, and four rib heights. Baseline values are provided for comparison. In general, at Re = 20,000, all cases showed higher local values of turbulence intensity than for the baseline, especially at the stagnation point where the maximum difference takes place, giving a percentage increase of approximately 60%. However, this was not the case for Re = 40,000, where local values of turbulence intensity between the stagnation point and the rib did not change with the presence of the rib until close to the front of the rib. This may be due to the inversely proportional relationship between turbulence intensity and flow speed. The value then dropped to zero and was followed by a substantial increase for every rib, due to before-rib recirculation vortices, the value of which depended on rib height. We see that for Re = 20,000, e/D = 0.056 is the rib height, which gives a maximum turbulence intensity of 0.54. At Re = 40,000, the maximum turbulence intensity was attained at rib height e/D = 0.037 with a turbulence intensity value of about 1. In both cases, e/D = 0.019 showed the minimum turbulence intensity. This was due to low levels of vortices before and after the rib.

b.Entropy Generation Analysis

Figure 9 illustrates the effect of changing rib height from (e/D) = 0.019 to 0.074 and changing the Reynolds number from 10,000 to 50,000 on entropy generation for a radial distance of R/D = 1.5. Generally, by increasing Re, both heat transfer and viscous dissipation increased, resulting in higher entropy generation due to heat.

For Re = 10,000 and 20,000, rib height e/D = 0.056 yielded the maximum $S_{h}$ of approximately 2940 J/K, while e = 0.019 produced a corresponding value of 2700 J/K. For Re ≥ 30,000, all the ribs produced levels of $S_{h}$ lower than for the no rib case and, in general terms, the higher the rib, the relatively lower the value of $S_{h}$ was. As for entropy generation due to friction $S_{f}$ for Re = 10,000, a rib height of 0.074 produced the minimum $S_{f}$ of about 99 J/K, while e/D = 0.019 gave a corresponding value of 150 J/K. For Re ≥ 40,000, the values of $S_{f}$ were greater than for the no rib case and, again, the higher the rib, then the lower the corresponding value of $S_{f}$ was. However, at a Reynolds number Re ≥ 40,000, the value of $S_{f}$ for the highest rib (0.074) fell below that of the no rib case.

In general, both $S_{h}$ and $S_{f}$ increased with the Reynolds number at rates depending on the amount of heat transferred and turbulence generated by the ribs. The rib of height e/D= 0.019 consistently produced the highest levels of entropy, though for $S_{h}\;$, this was below the no rib value for Re that was greater than approximately 20,000.

Figure 10 shows the effect of changing the rib height and the Reynolds number from 20,000 to 40,000 on local values of total entropy, compared to the no rib case. For the lower value of Re, all local values (no rib included) of total entropy generation had the same magnitude at the stagnation point (R/D = 0), whereas for Re = 40,000, all rib cases showed a lower value than for the no rib case. In general, the higher the Reynolds number, the bigger the size of the before-rib recirculation zone, and the lower the values of total entropy generation were.

All the plots showed a drop in *S_t_* immediately before the rib, and then a sudden and dramatic rise, due to the existence of the flow recirculation zone before the rib. When the flow met the rib, a huge peak in the rate of heat transfer occurred, resulting in a peak value in *S_t_*. The peak value varied with both rib height and Reynolds number. For Re = 20,000, e/D = 0.019 gave the maximum peak value of $S_{t}$, about 340% more than the no rib case, and about 4.3% than e/D = 0.037. However, for Re = 40,000, the maximum peak value of $S_{t}$ was for e/D = 0.037, with a percentage increase of approximately 20% more than for e/D = 0.019 at the same radial location. The panels in the Figure present the contours for $S_{t},$ showing the precise locations of the peak value for the four rib heights on the top leading edge of the rib, because of the direct contact between the moving flow and the rib.

The Bejan number is the ratio of entropy generated due to heat exchange to the total entropy generated. The Bejan number for a smooth wall and wall with four ribs of different heights and a range of Reynolds numbers is shown in Figure 11. In general, for a fixed rib location, the Bejan number decreased as the Re increased and also increased as the rib height increased, which concurs with the result reported by Alhajeri et al. [36]. This is due to the combination of both flow velocity and before-rib recirculation zone on both *S_h_* and *S_t_*. This is another way of showing that turbulent friction between the rib and the flow results in relatively lower convective heat transfer, which then decreases entropy generation due to heat exchange, relative to entropy generated due to friction.

### 3.3. Roughened Wall—Rib Location Effect

Flow Physics Analysis

Figure 12 compares wall shear stress distribution for the no rib case and with the rib at a height of R/D = 0.037 and a square cross-section. The Reynolds numbers were 10,000, 30,000 and 50,000 for each of the three rib locations. Generally, regardless of its location, the rib had only a minor effect on wall shear from the stagnation point to a radial distance close to where it was located. The rib’s effect occurred just when the flow reached the rib, causing the first local minima due to the effect of the small flow recirculation zone on the pressure gradient (*∂P*/*∂y*). When the jet hit the rib, a wall shear peak occurred causing it to separate from the wall, forming a low-pressure region where a larger flow recirculation zone occurred, which affected the main flow velocity, creating a local maximum and second local minimum value of shear stress. The contour shown in the Figure displays a maximum wall shear peak location on the rib. The magnitude of the peak wall shear stress decreased as the rib location moved downstream, as does the stagnation point, due to the momentum exchange between the flow and the heated wall. The peak value of wall shear stress at R/D = 1 was approximately 40% bigger than its value at R/D = 2.

To continue examining the effect of rib location on flow kinematics, Figure 13 illustrates the variation of turbulence intensity in the radial direction along the heated plate. For comparison, the turbulence intensity for the no rib case is also shown. As expected, the major effect on turbulence intensity occurred when the flow reached the rib. This effect took the form of an increase in intensity due to the collision between the flow and the rib, resulting in 550%, 760%, and 440% percentage increases for Re = 10,000, 20,000, and 30,000, respectively, when the rib was at R/D = 1. The Figure also shows that as Re increased, the peak value of turbulence intensity increased, but the further the rib was located downstream, the more the peak value decreased, due to the momentum exchange between the flow and the wall. In the case of Re = 30,000, the peak value decreased by about 12% and 41% for R/D = 1.5 and 2, respectively, when compared to R/D = 1. The turbulence intensity contour shows the location of this peak on the rib itself, at the rib edge, where the flow directly hit the rib, and where the velocity streamlines demonstrated how vortices prevent the flow from directly colliding with the rib, resulting in reduced turbulence intensity. Briefly, the rib location and the Reynolds number had a noticeable effect on turbulence intensity, as both are related to flow velocity.

b.Entropy Generation Analysis

Figure 14 represents the heat and friction entropy generation for variable Reynolds numbers and rib radial locations and compares it to the baseline case. The Figure demonstrates the effect of changing rib radial location from R/D = 1 to R/D = 2 on both types of entropy generation, under the influence of 10,000 ≤ Re ≤ 50,000 for a specific rib height e/D = 0.037. The results show that for rib locations (R/D) = 1 and 1.5, the values of ($S_{h}$) were higher when compared to the baseline case. This is true only for certain values of Re, depending on the rib location. For example, placing the rib at R/D = 1 gave a higher ($S_{h}$) than the baseline case, until Re ≤ 40,000, when these values started to decay more than the no rib case, as the Re increased. All these findings may be due to the effects of both rib location and flow velocity, as they both contribute to forming a before-rib recirculation zone on the heat transfer rate between the flow and the wall. However, all ($S_{h}$) values of R/D = 2 were either matched values of the baseline case for Re ≤ 20,000, or less than that for Re ≥ 20,000. This can be ascribed to the high turbulence, as both the Re and the rib location increased. As for entropy generation due to friction $S_{f}$, the complicity of the flow in jet impingement did not seem to play a big role, as all curves showed a clear observation where friction was minimal for the no rib case and maximum for R/D = 1 for all Re values. The $S_{f}$ values, when compared to the no rib case, increased as Re increased and R/D decreased due to various wall frictions, which depended on both the Re and rib location. All previous findings led to the fact that placing the rib at R/D = 1 yielded the highest total entropy generation ($S_{t}$). In general, varying the rib location and Re values had a noticeable impact on $S_{h}$, $S_{f}$ and ($S_{t}$).

Figure 15 shows velocity streamlines and contours of $S_{h}$, $S_{f}$ and $S_{t}$ for e/D = 0.037, Re = 30,000 and different R/D. The Figure illustrates the effect of changing the rib location from R/D = 1 to R/D = 2 on the two main types of entropy generation resulting from a cold jet impinging on a heated roughened wall. The Figure shows that by changing the rib location, the local values of $S_{h}$ and $S_{f}$ noticeably changed. For example, the average value of $S_{h}$ for R/D = 1 was approximately 30% bigger than it was for R/D = 2, as R/D = 1 was about 55% bigger than it was for R/D = 2. This was due to the enhancement of heat transfer exchange between the heated wall and the cold fluid because of the high turbulence level induced by the flow recirculation before and after the rib, for each rib location. The Figure also demonstrates that the maximum local values of $S_{h}$ and $S_{f}$ occurred on the two-right edges of the rib where the main jet directly hits the protrusion as the continued flow recirculation prevented the flow from directly hitting the whole right side of the rib. In general, rib location R/D = 1 gave the maximum total entropy generation when compared to other rib locations due to the fact that, placing the rib at a radial distance R = 1 D meant placing it within the stagnation region, where high turbulence intensity existed.

Figure 16 indicates that a flow separation occurred in front of the rib, resulting in a small separation region followed by a larger after-rib recirculation vortex. The Figure indicates that entropy generation had high values around the roughening element, especially after the rib. This was attributed to large vortices which resulted in higher shear and temperature gradient in this region. It was observed that a small recirculation region occurred on top of the rib, causing a continued fresh boundary layer, which increased heat transfer in this particular small area, which consensually increased total entropy generation.

The Bejan number calculated from Equation (14) is shown in Figure 17. The Figure demonstrates the Bejan number for Reynolds numbers between 10,000 and 50,000, and rib locations between 1 and 2, and compares it to the no rib case. Generally, the Bejan number decreased almost linearly as Re increased, due to higher convective heat transfer and friction between the main jet and the moving flow. The Figure also demonstrates that the Bejan number increased as the R/D increased for all tested Reynolds numbers, giving higher convective heat transfer.

## 4. Conclusions

This paper has numerically studied how the presence of surface roughness can impact both flow physics and entropy generated, using the commercial software ANSYS V 18.1 system. This impact was investigated and compared for a smooth, heated, horizontal plate and surface roughness was added to it in the shape of a rib with a square cross-section. The rib radius was at R = 1, 1.5 and 2 D, where the square cross-section varied from 0.019 to 0.074 in steps of 0.019. The jet Reynolds number was changed from 10,000 to 50,000.

A rib height of e/D = 0.019 produced the smallest size of a before-rib flow recirculation zone, preventing direct contact between the flow and the wall, resulting in the greatest wall shear stress peak value, but the lowest turbulence intensity.A rib height of e/D = 0.019 had a minor contribution to enhancing the heat transfer rate between the flow and the heated wall, due to the low level of flow vortices occurring by this rib height wall, resulting in the minimum *S_h_*.At higher Reynolds numbers, adding a rib of a suitable height could have reduced the total entropy because it reduces *S_h_* more than it increases *S_f_*. In addition, for ribs of heights 0.019, 0.037 and 0.056, the entropy generated by friction was greater than that due to heat exchange.In general, varying the rib location and Re values had a noticeable impact on $S_{h}$, $S_{f}$ and ($S_{t}$), as these variations play an important role in forming and sizing a before-rib recirculation zone.Placing the rib at a radial location R/D = 1 involved positioning it within the stagnation region where the flow was very active, causing higher flow-wall friction and flow recirculation. This maximized $S_{f}$ but minimized $S_{t}$, when compared to other radial locations.In general, the Bejan number decreased as the Re increased. The rib height and rib location decreased due to the combined effect of three parameters: turbulence intensity, flow recirculation and wall friction, on total entropy generation. The results shown here could have important practical applications in the design of turbine blades, nozzle guide vanes and electronic devices for maximum heat removal.

For further research, the importance of the shape of the roughening element and jet impingement angle on entropy generation should be investigated.

## Figures and Tables

**Figure 1 entropy-24-00661-f001:**
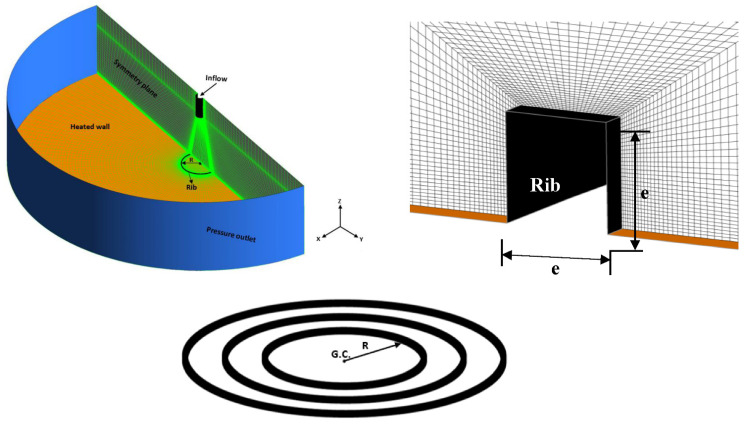
Computational domain and grid topology.

**Figure 2 entropy-24-00661-f002:**
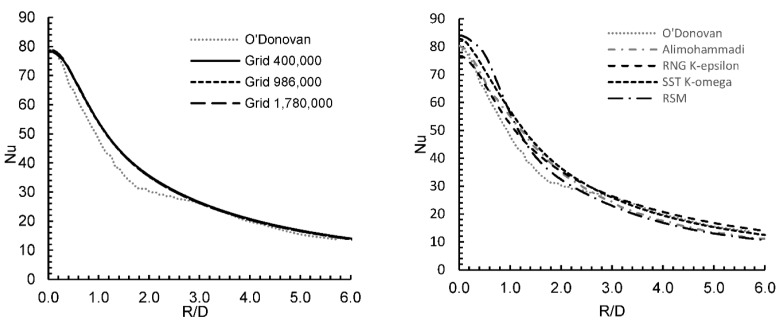
Model validation of smooth wall. O’Donovan, T.S. et al. [32], Alimohammadi, S.; et al. [35].

**Figure 3 entropy-24-00661-f003:**
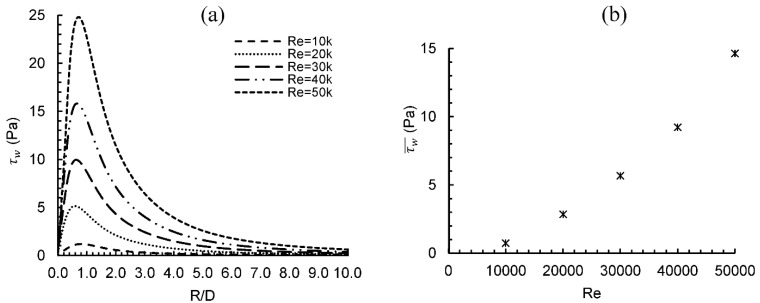
Wall shear stress distribution for different Re numbers, (**a**) local values, (**b**) average values.

**Figure 4 entropy-24-00661-f004:**
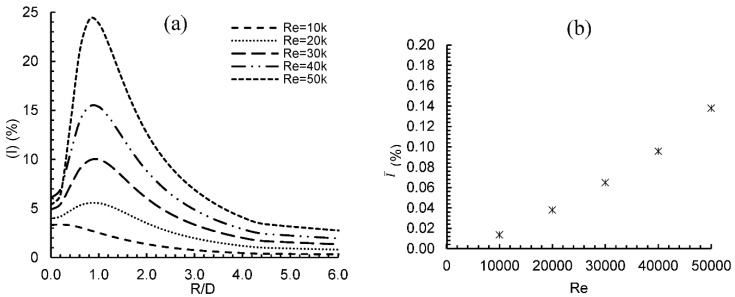
Turbulence intensity distribution: (**a**) Local I over the flat plate, (**b**) Average I for variable Re values.

**Figure 5 entropy-24-00661-f005:**
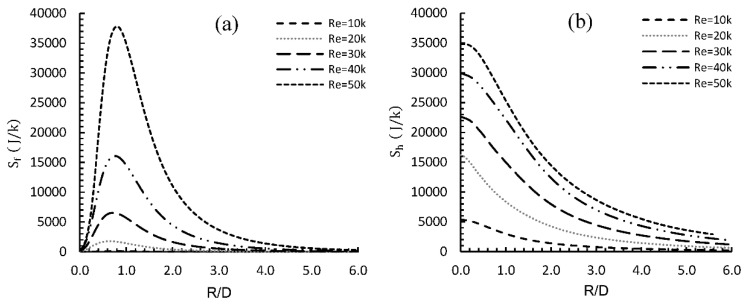
(**a**) Entropy generation due to friction (*S_f_*) and (**b**) due to heat (*S_h_*) for different Re at different radial locations (R/D).

**Figure 6 entropy-24-00661-f006:**
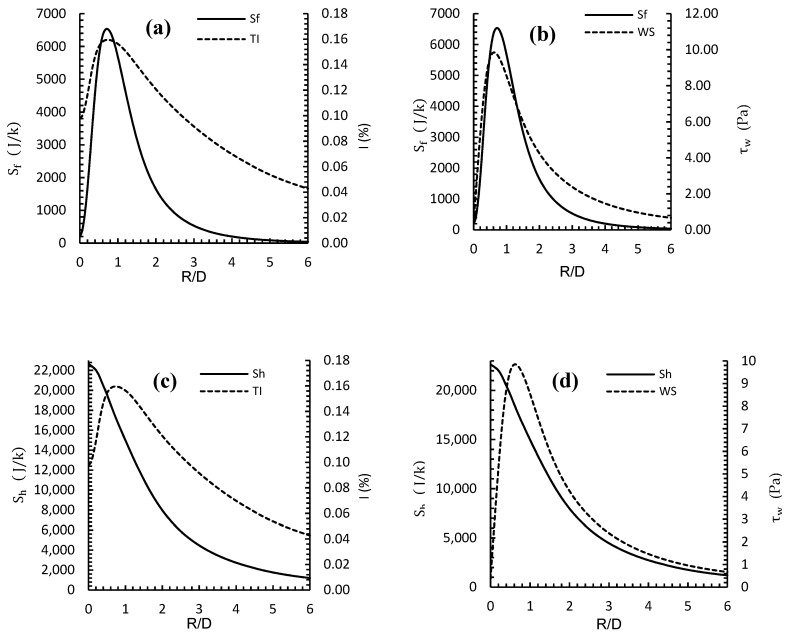
Relationship between (**a**) (*S_f_*) (**b**) (*S_h_*) and (**c**) flow turbulence intensity (I) and (**d**) wall shear (*τ_w_*) at different radial distances (R/D) at Re = 30,000 with a smooth wall.

**Figure 7 entropy-24-00661-f007:**
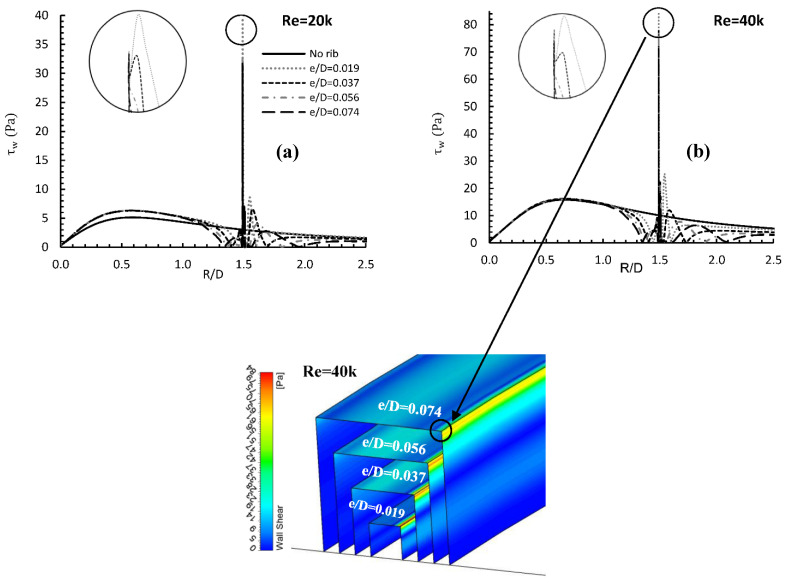
Contour and local wall shear distribution and contours for (**a**) Re = 20,000 and (**b**) 40,000, and three rib heights.

**Figure 8 entropy-24-00661-f008:**
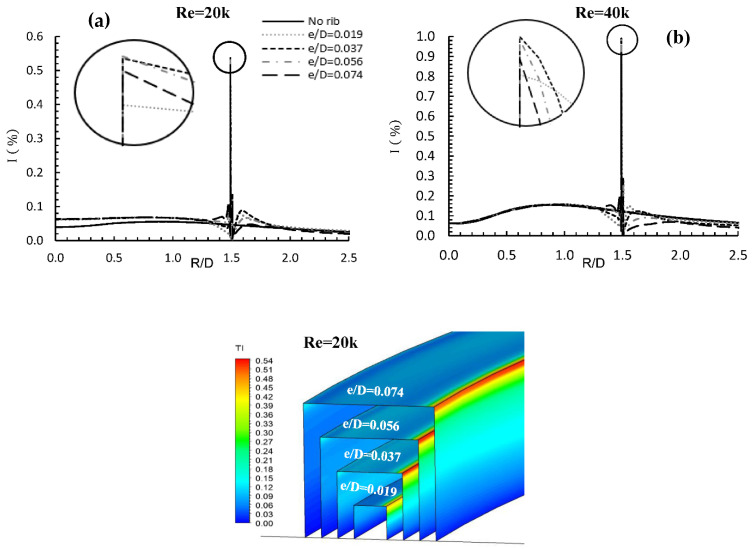
Turbulence intensity for R/D = 1.5, two flow speeds (**a**) Re = 10,000 and (**b**) Re = 20,000 and four rib heights (e).

**Figure 9 entropy-24-00661-f009:**
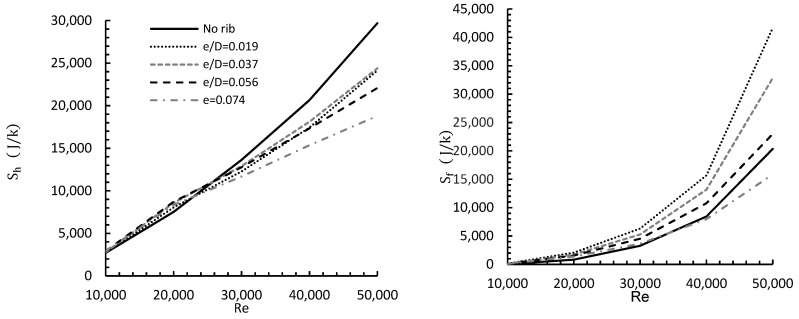
Effect of varying rib height and Reynolds number on different entropy types for R/D = 1.5.

**Figure 10 entropy-24-00661-f010:**
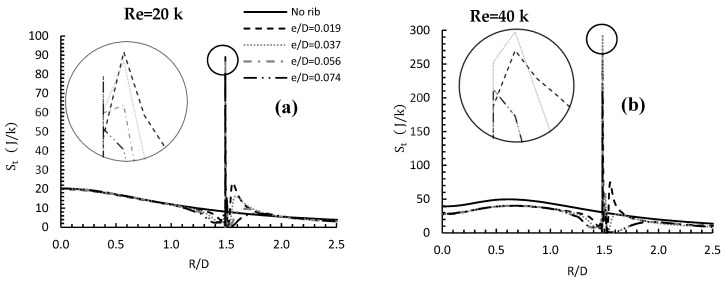
Total entropy generation and velocity streamlines contours for R/D = 1.5, for (**a**) Re = 20,000 and (**b**) Re = 40,000 for different rib heights.

**Figure 11 entropy-24-00661-f011:**
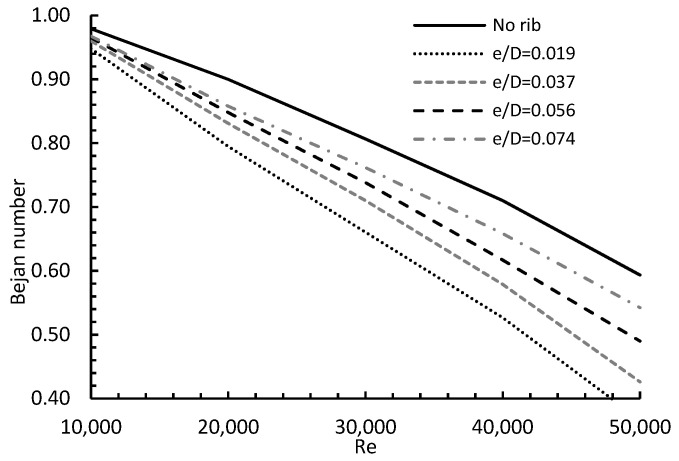
Bejan number for R/D = 1.5, different e and different Re.

**Figure 12 entropy-24-00661-f012:**
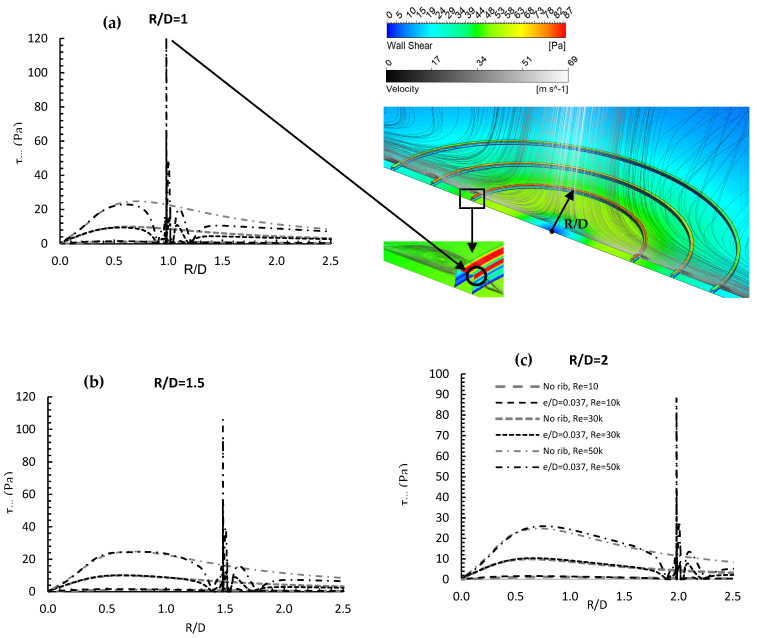
Wall shear distribution and velocity streamlines in the radial direction for e/D = 0.037, Re = 30,000 for (**a**) R/D = 1, (**b**) R/D = 1.5, and (**c**) R/D = 2.

**Figure 13 entropy-24-00661-f013:**
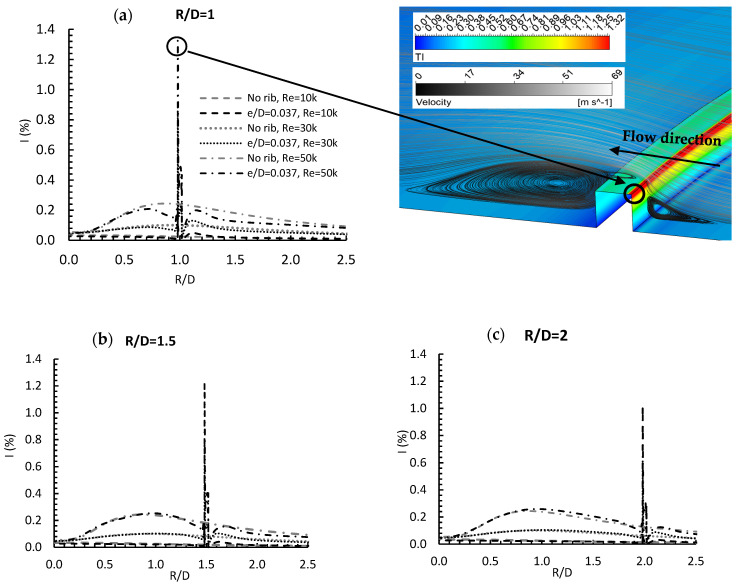
Turbulence intensity distribution and velocity streamlines and in radial direction for e/D = 0.037, Re = 30,000 for (**a**) R/D = 1, (**b**) R/D = 1.5, and (**c**) R/D = 2.

**Figure 14 entropy-24-00661-f014:**
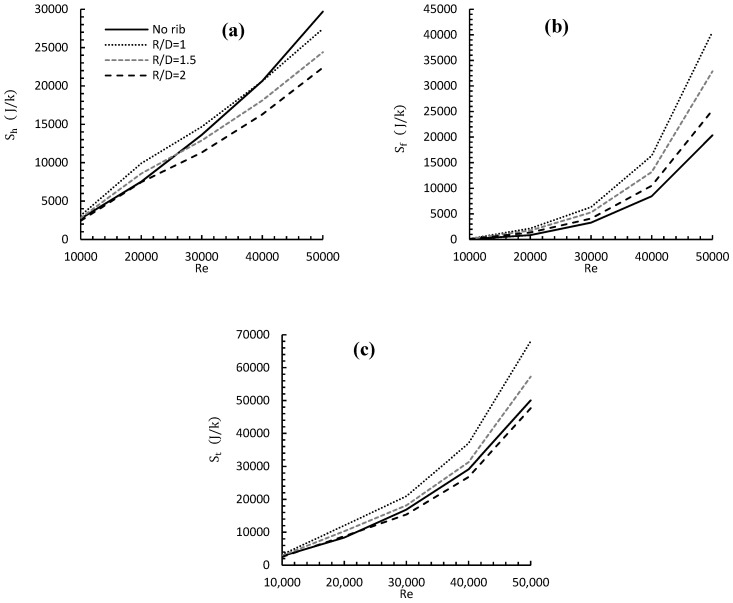
Effect of varying rib radial location and Reynolds number on (**a**) entropy due to heat, (**b**) entropy due to friction, and (**c**) total entropy for e/D = 0.037.

**Figure 15 entropy-24-00661-f015:**
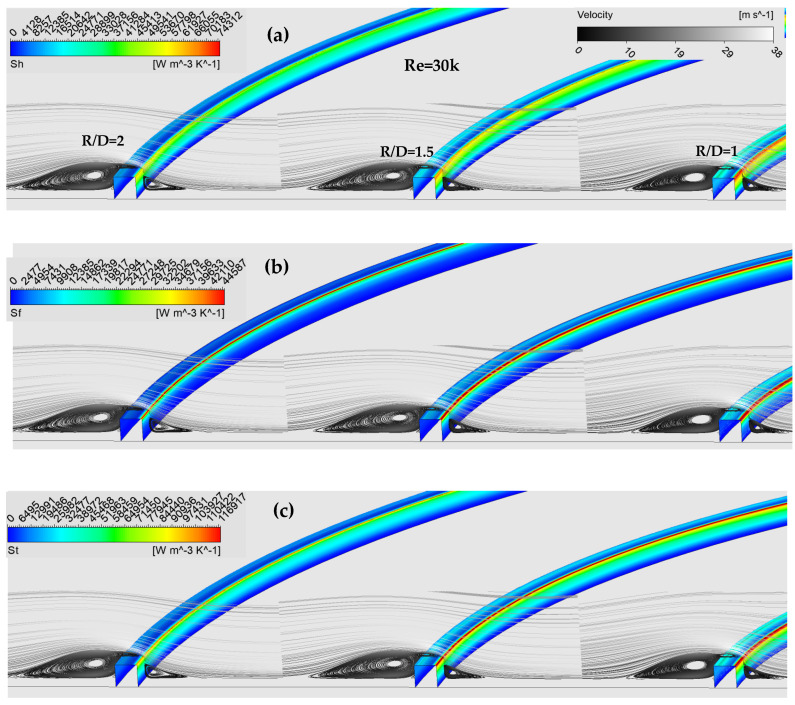
Velocity streamlines and contours of (**a**) *S_h_*, (**b**) *S_f_* and (**c**) *S_t_* for e/D = 0.037, Re = 30,000 and different R/D.

**Figure 16 entropy-24-00661-f016:**
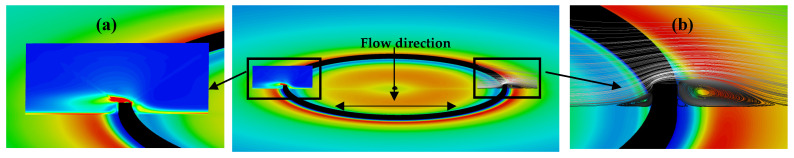
(**a**) Total entropy contours and (**b**) velocity streamlines of e/D = 0.037, Re = 30,000 and R/D = 1.

**Figure 17 entropy-24-00661-f017:**
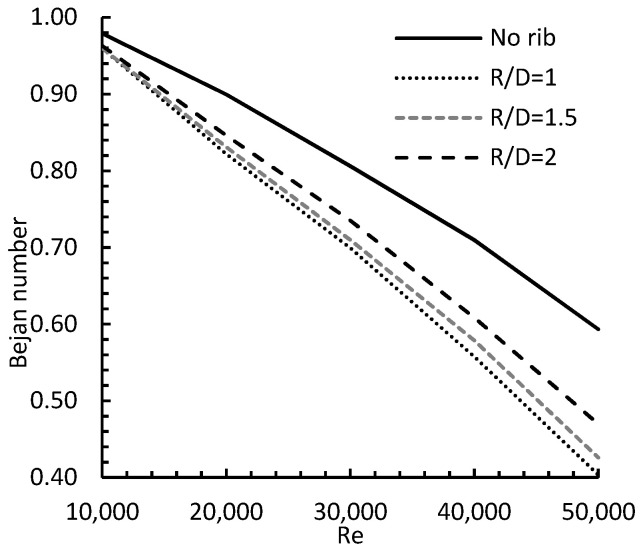
Bejan number for e/D = 0.037, different R/D and different Re.

**Table 1 entropy-24-00661-t001:** Air properties at ambient temperature.

Air Properties	Value
Density (ρ)	1.177 kg/m^3^
Dynamic viscosity (μ)	1.846 × 10^−5^ kg/m s
Thermal conductivity (k)	0.026 W/m K
Specific heat (*C_p_*)	1004.9 kJ/kg K
Corresponding Prandtl no. (*P_r_*)	0.0706

## Data Availability

Not applicable.

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
