# Peer review of "Impact of Surface Roughness on Flow Physics and Entropy Generation in Jet Impingement Applications"

_entropy, 2022, doi:10.3390/e24050661_

Round 1

Reviewer 1 Report

Some specific comments:

1) I think I have mentioned the necessity of meticulous proofreading and English editing (grammar, spelling, punctuation, etc.) in the list of comments I had already provided, but I am going to emphasize it again.

2) I suggest sticking to the normalized parameters. For example, you present the radius as R/D with D being the jet's diameter, but then the roughness element height is given as "e" in mm.

3) The authors did not mention what characteristic length they used for computing the Reynolds numbers. Was it based on the jet's dimensions or the roughness element? I cannot make sense of the Reynolds number unless I know how exactly it is calculated (what dimension was used? Did that dimension change between different cases? What velocity was used? Was it the jet velocity? Did you change the velocity? How did you change the Reynolds number between 10,000 and 50,000? Via length or by velocity? These are important questions.)

4) The governing equation section misses the turbulence model.

5) The governing equation section needs to define every variable. For example, what is the difference between V and v', T and t', etc.? What are these parameters? You know they are mean and fluctuating terms, but not every reader knows that, and it is standard practice to define every variable.

6) The definition provided in lines 209 and 210 for the Bejan number is wrong (a typo?). Bejan's number is not an energy ratio but an entropy ratio.

General comment:

The main issue I notice in this article is the way the results are being discussed. The authors do not provide a high-level physics-based discussion that teaches us something important. I am not saying there is nothing important in the results; I am saying that is not provided in the discussion and the conclusion. Rather than that, the authors provide a detailed report of all the numbers and trends we observe in the figures.

For example, a bullet point provided in the conclusion section reads as:

"A rib of height of e= 0.25 mm produced the greatest wall shear stress peak value but lowest turbulence intensity due to the low level of vortex generation where a rib of height 1.00 mm produced the lowest peak value."

What are we supposed to learn from this? Instead, I would like to see a more general discussion (of course, based on the data presented in the paper) explaining how and "why" the rib height affects turbulence intensity and wall shear stress (in general). I also would like to see a physics-based discussion on whether this general discussion would hold within other Reynolds number regimes? You can simply say you do not know that, but you can share your thoughts if there is anything you know. Ok, all I am saying is that the way the discussion and the conclusion are drafted is exceptionally boring, and it is tough for the readers to extract any general conclusions and learning outcomes.

Reviewer 2 Report

This is well-written article and I am glad to accept the paper in present form.

Author Response

Dear Reviewer

Greetings

First of all, I would like to thank you for your recommendations to publish the paper as it is. I appreciate your effort .

With kind regards

Reviewer 3 Report

The manuscript analyzes the impact of artificial surface roughness on vortex generation and heat transfer modulation at the expense of increased entropy generation. The authors explore numerically the flow physics and entropy generation promoted by synthetic surface roughness.

The results by the authors are innovative and could be valuable in the development and integration of synthetic surface roughness. In this line, I believe this manuscript could be suitable for publication. However, before recommending its publication I suggest the authors address the following items in an attempt to boost its outreach and improve the presentation of the results.

Review the sentence starting in line 44.

The first and second sentences in the introduction are completely unconnected, the narrative in the introduction must be improved.

The cross-references are wrongly implemented which impedes the correct lecture of the text.

The authors should describe how are the air properties being modeled, are they kept constant? The consequences of the modeling approach, advantages, and limitations should also be exposed.

Did the authors monitor any integral quantity, such as overall shear stress or mass flow to ensure convergence? Numerical convergence based on residuals is usually unreliable, as residuals only compare numerical evolution but not physical predictions.

The authors use a validation case based on the Nusselt number distribution. However, the overall discussion of the results is based on entropy evolution. To further support the results the authors should also include a validation case comparing the entropy generation.

Author Response

This manuscript is a resubmission of an earlier submission. The following is a list of the peer review reports and author responses from that submission.